# IS MARGIN ALL YOU NEED? AN EXTENSIVE EMPIRICAL STUDY OF DEEP ACTIVE LEARNING ON TABULAR DATA

## ABSTRACT

Given a labeled training set and a collection of unlabeled data, the goal of active learning (AL) is to identify the best unlabeled points to label. In this comprehensive study, we analyze the performance of a variety of AL algorithms on deep neural networks trained on 69 real-world tabular classification datasets from the OpenML-CC18 benchmark. We consider different data regimes and the effect of self-supervised model pre-training. Surprisingly, we find that the classical margin sampling technique matches or outperforms all others, including current state-of-art, in a wide range of experimental settings. To researchers, we hope to encourage rigorous benchmarking against margin, and to practitioners facing tabular data labeling constraints that hyper-parameter-free margin may often be all they need.

## 1 INTRODUCTION

Active learning (AL), the problem of identifying examples to label, is an important problem in machine learning since obtaining labels for data is oftentimes a costly manual process. Being able to efficiently select which points to label can reduce the cost of model learning tremendously. High-quality data is a key component in any machine learning system and has a very large influence on the results of that system (Cortes et al., 1994; Gudivada et al., 2017; Willemink et al., 2020); thus, improving data curation can potentially be fruitful for the entire ML pipeline.

Margin sampling, also referred to as uncertainty sampling (Lewis et al., 1996; MacKay, 1992), is a classical active learning technique that chooses the classifier's most uncertain examples to label. In the context of modern deep neural networks, the margin method scores each example by the difference between the top two confidence (e.g. softmax) scores of the model's prediction. In practical and industrial settings, margin is used extensively in a wide range of areas including computational drug discovery (Reker & Schneider, 2015; Warmuth et al., 2001), magnetic resonance imaging (Liebgott et al., 2016), named entity recognition (Shen et al., 2017), as well as predictive models for weather (Chen et al., 2012), autonomous driving (Hussein et al., 2016), network traffic (Shahraki et al., 2021), and financial fraud prediction (Karlos et al., 2017).

Since the margin sampling method is very simple, it seems particularly appealing to try to modify and improve on it, or even develop more complex AL methods to replace it. Indeed, many papers in the literature have proposed such methods that, at least in the particular settings considered, consistently outperform margin. In this paper, we put this intuition to the test by doing a head-to-head comparison of margin with a number of recently proposed state-of-the-art active learning methods across a variety of tabular datasets. We show that in the end, margin matches or outperforms all other methods consistently in almost all situations. Thus, our results suggest that practitioners of active learning working with tabular datasets, similar to the ones we consider here, should keep things simple and stick to the good old margin method.

In many previous AL studies, the improvements over margin are oftentimes only in settings that are not representative of all practical use cases. One such scenario is the large-batch case, where the number of examples to be labeled at once is large. It is often argued that margin is not the optimal strategy in this situation because it exhausts the labeling budget on a very narrow set of points close to decision boundary of the model and introducing more diversity would have helped (Huo & Tang, 2014; Sener & Savarese, 2017; Cai et al., 2021). However, some studies find that the number of

examples to be labeled at once has to be very high before there is advantage over margin (Jiang & Gupta, 2021) and in practice a large batch of examples usually does not need to be labeled at once, and it is to the learners' advantage to use smaller batch sizes so that as datapoints get labeled, such information can be incorporated to re-train the model and thus choose the next examples in a more informed way. It is important to point out, however, that in some cases, re-training the model is very costly (Citovsky et al., 2021). In that case, gathering a larger batch could be beneficial. In this study, we focus on the practically more common setting of AL that allows frequent retraining of the model.

Many papers also only restrict their study to only a couple of benchmark datasets and while such proposals may outperform margin, these results don't necessarily carry over to a broader set of datasets and thus such studies may have the unintended consequence of overfitting to the dataset.

In the real world live active learning setting, examples are sent to human labelers and thus we don't have the luxury of comparing multiple active learning methods or even tuning the hyper-parameters of a single method, without incurring significantly higher labeling cost. Instead, we have to commit to a single active learning method oftentimes without much information. Our results on the OpenML-CC18 benchmark suggest that in almost all cases when training with tabular data, it is safe for practitioners to commit to margin sampling (which comes with the welcome property of not having additional hyper-parameters) and have the peace of mind that other alternatives wouldn't have performed better in a statistically significant way.

## 2    RELATED WORK

There have been a number of works in the literature providing an empirical analysis of active learning procedures in the context of tabular data. Schein & Ungar (2007) studies active learning procedures for logistic regression and show that margin sampling performs most favorably. Ramirez-Loaiza et al. (2017) show that with simple datasets and models, margin sampling performs better compared to random and Query-by-Committee if accuracy is the metric, while they found random performs best under the AUC metric. Pereira-Santos et al. (2019) provides a investigation of the performance of active learning strategies with various models including SVMs, random forests, and nearest neighbors. They find that using margin with random forests was the strongest combination. Our study focuses on the accuracy metric and also shows that margin is the strongest baseline, but is much more relevant to the modern deep learning setting and with a comparison to a much more expanded set of baselines and datasets. Our focus on neural networks is timely as recent work (Bahri et al., 2021) showed that neural networks often outperform traditional approaches for modeling tabular data, like Gradient Boosted Decision Trees (Chen & Guestrin, 2016), particularly when they are pre-trained in the way we explore here. To our knowledge we provide the most comprehensive and practically relevant empirical study of active learning baselines on neural networks thus far.

There have also been empirical evaluations of active learning procedures in the non-tabular case. Hu et al. (2021) showed that margin attained the best average performance of the baselines tested on two image and three text classification tasks across a variety of neural network architectures and labeling budgets. Munjal et al. (2022) showed that on the image classification benchmarks CIFAR-10, CIFAR-100, and ImageNet, under strong regularization, none of the numerous active learning baselines they tested had a meaningful advantage over random sampling. We hypothesize that this may be due to the initial network having too little information (i.e. no pre-training and small initial seed set) for active learning to be effective and conclusions may be different otherwise. It is also worth noting that many active learning studies in computer vision only present results on a few benchmark datasets (Munjal et al., 2022; Sener & Savarese, 2017; Beluch et al., 2018; Emam et al., 2021; Mottaghi & Yeung, 2019; Hu et al., 2018), and while they may have promising results on such datasets, it's unclear how they translate to a wider set of computer vision datasets. We show that many of these ideas do not perform well when put to the test on our extensive tabular dataset setting. Dor et al. (2020) evaluated various active learning baselines for BERT and showed in most cases, margin provided the most statistically significant advantage over passive learning. One useful direction for future work is establishing an extensive empirical study for computer vision and NLP.

While our study is empirical, it is worth mentioning that despite being such a simple and classical baseline, margin is difficult to analyze theoretically and there remains little theoretical understanding of the method. Balcan et al. (2007) provides learning bounds for a modification of margin where examples are labeled in batches where the batch sizes depend on predetermined thresholds and

assume that the data is drawn on the unit ball and the set of classifiers are the linear separators on the ball that pass through the center of the ball (i.e. no bias term). Wang & Singh (2016) provide a noise-adaptive extension under this style of analysis. Recently, Raj & Bach (2022) proposed a general family of margin-based active learning procedures for SGD-based learners that comes with theoretical guarantees; however these algorithm require a predetermined sampling function and hence, like the previous works, does not provide any guarantees for the classical margin procedure with a fixed batch size. Without such theoretical understanding of popular active learning procedures, having comprehensive and robust empirical studies become even more important for our understanding of active learning.

## 3 PROBLEM STATEMENT

We begin with a brief overview of pool-based active learning. One has an initial sample of $S$ labeled training examples $\mathcal{I}_0 = \{(x_i, y_i)\}_{i=1}^S$ where $x_i \in \mathbb{R}^D$ and $y_i \in \mathbb{N}$. We also assume a pool of unlabeled examples $\mathcal{Z} = \{z \in \mathbb{R}^D\}$ that can be selected for labeling, for a grand total of $N_{\text{total}}$ points. The goal is for rounds $t = 1, 2, ..., T$, to select $B$ examples (active learning batch size) in each round from $\mathcal{Z}$ to be labeled and added to the train set $\mathcal{I}_{t-1}$ (and removed from $\mathcal{Z}$) to produce the new labeled set $\mathcal{I}_t$ .

### 3.1 SCARF

Recently, Bahri et al. (2021) proposed a technique for pre-training deep networks on tabular datasets they term SCARF. Leveraging self-supervised contrastive learning in a spirit similar to vision-oriented SimCLR (Chen et al., 2020), the method teaches the model to embed an example $x_i$ and its corrupted view $\tilde{x}_i$ closer together in space than $x_i$ and $\tilde{x}_j$, the corrupted view of a different point $x_j$. They show that pre-training boosts test accuracy on the same benchmark datasets we consider here even when the labels are noisy, and even when labeled data is limited and semi-supervised methods are used. We investigate the effect of SCARF pre-training on AL methods.

### 3.2 ACTIVE LEARNING BASELINES

**Margin, Entropy, Least Confident.** These three popular methods score candidates using the uncertainty of a single model, $p$. It is assumed that at round $t$, $p$ is trained on the labeled set thus far to select the next batch of examples with highest uncertainty scores in $\mathcal{Z}$. We seek points with the smallest margin, largest entropy, or largest least confident (LC) scores, defined as follows:

$$\text{Margin}(x) = p(y = y_1(x)|x) - p(y = y_2(x)|x), \text{ where}$$
$$y_1(x) = \arg\max_c p(y = c|x),$$
$$y_2(x) = \arg\max_{c|c \neq y_1(x)} p(y = c|x).$$
$$\text{Entropy}(x) = -\sum_c p(y = c|x) \log p(y = c|x).$$
$$\text{LC}(x) = 1 - \max_c p(y = c|x).$$

**Random-Margin.** A 50-50 mix of random and margin; a common way to enhance diversity. Half of the batch is chosen based on margin, and the other half of the examples are randomly selected.

**Min-Margin** (Jiang & Gupta, 2021). An extension of margin that uses bootstrapped models to increase the diversity of the chosen batch. $K = 25$ models are trained on bootstrap samples drawn from the active set (where the bootstrap is done on a per-class basis with the sample size the same as the original training dataset size), and the minimum margin across the $K$ models is used as the score.

**Typical Clustering (TypiClust)** (Hacohen et al., 2022). A method that uses self-supervised embeddings to balance selection of "typical" or representative points with diverse ones as follows. At round $t$, all $N_{\text{total}}$ pre-trained embeddings are clustered into $|\mathcal{I}_{t-1}| + B$ clusters using $K$-means and then the most typical examples from the $B$ largest uncovered clusters (i.e. clusters containing no points from $\mathcal{I}_{i-1}$) are selected. Given that cosine similarity is the distance used when learning SCARF

embeddings, we use spherical $K$-means and define the typicality score as:

$$\text{Typicality}(x) = \left( \frac{1}{k} \sum_{x_i \in \text{k-NN}(x)} \frac{1 - \text{CosSim}(x, x_i)}{2} \right)^{-1}.$$

$k$ is chosen as $\min\{N_{\text{total}}, \max\{20, |C(x)|\}\}$, where $|C(x)|$ is the size of the cluster containing $x$.

**Maximum Entropy (MaxEnt) and BALD.** These Bayesian-based approaches use $M$ models drawn from a posterior. Oftentimes, Monte-Carlo dropout (MC-dropout) is used, wherein a single model is trained with dropout and then $M$ different dropout masks are sampled and applied during inference (Gal et al., 2017). This can be seen as model inference using different models with weights $\{w_m\}_{i=1}^{M}$. For maximum entropy, we score using the model's entropy $H$:

$$H[y|x] = -\sum_c \alpha_c(x) \log \alpha_c(x), \text{ where}$$

$$\alpha_c(x) = \frac{1}{M} \sum_{m=1}^{M} p(y = c|x, w_m).$$

BALD (Houlsby et al., 2011) estimates the mutual information (MI) between the datapoints and the model weights, the idea being that points with large MI between the predicted label and weights have a larger impact on the trained model's performance. The measure, denoted $I$, is approximated as:

$$I[y|x] = H[y|x] - \frac{1}{M} \sum_{m=1}^{M} \sum_c -\beta_{c,m}(x) \log \beta_{c,m}(x), \text{ where}$$

$$\beta_{c,m}(x) = p(y = c|x, w_m).$$

We use $M = 25$ and a dropout rate of $0.5$ as done in prior work (Beluch et al., 2018). For consistency, dropout is not applied during pre-training, only fine-tuning.

**BADGE** (Ash et al., 2019). Batch Active learning by Diverse Gradient Embeddings (BADGE) uses the loss gradient of the neural network's final dense layer for each unlabeled sample, where the loss is computed using the model's most likely label for the sample. The gradient embeddings are clustered using the $K$-means++ seeding algorithm (Arthur & Vassilvitskii, 2006) and the centroids are the samples added to the labeled set. The gradient embedding for a sample captures how uncertain the model is about the sample's label while the clustering provides diversity in sample selection. We use the `sklearn.cluster.kmeans_plusplus` function with the default settings.

**CoreSet** (Sener & Savarese, 2017). Selects points to optimally cover the samples in embedding space. Specifically, at each acquisition round, it grows the active set one sample at a time for $B$ iterations. In each iteration, the candidate point $x_i$ that maximizes the distance between itself and its closest neighbor $x_j$ in the current active set is added. We use Euclidean distance on the activations of the penultimate layer (the layer immediately before the classification head), as done in Citovsky et al. (2021).

**Margin-Density** (Nguyen & Smeulders, 2004). Scores candidates by the product of their margin and their density estimates, so as to increase diversity. The density is computed by first clustering the penultimate layer activations of the current model on all $|\mathcal{Z}|$ candidate points via $K$-means. Then, the density score of candidate $x_i$ is computed as: $|C(x_i)|/|\mathcal{Z}|$, where $C(x_i)$ is the cluster containing $x_i$. We use $\min\{20, |\mathcal{Z}|\}$ clusters.

**Cluster-Margin** (Citovsky et al., 2021). Designed as a way to increase diversity in the extremely large batch size (100K-1M) setting where continuously model retraining can be expensive, Cluster-Margin prescribes a two step procedure. First, after the model is trained on the seed set, penultimate layer activations for all points are extracted and clustered using agglomerative clustering. This clustering is done only once. During each acquisition round, candidates with the $m \times B$ lowest margin scores (denoted $M$) along with their clusters $C_M$ are retrieved. $C_M$ is sorted ascendingly by cluster size and cycled through in order, selecting a single example at random from each cluster. After sampling from the final (i.e. largest) cluster, points are repeatedly sampled from the smallest unsaturated cluster until a total of $B$ points have been acquired. We explore the same settings as they do: $m = 1.25$ as well as $m = 10$. We use Scikit-Learn's (Pedregosa et al., 2011) agglomerative clustering API with Euclidean distance, average linkage, and number of clusters set to $\lfloor N_{\text{total}}/m \rfloor$.

**Query-by-Committee (QBC)** Beluch et al. (2018). Uses the disagreement among models in an ensemble, or committee, of $K$ models trained on the active set at each iteration. Like Munjal et al. (2022), we use the *variance ratio* $v$ which was shown to give superior results. It is computed as $v = 1 - f_m/K$, where $f_m$ is the number of predictions in the modal class. We set $K = 25$. As noted in prior work, differences among the committee members largely stem from differences in random initialization of the model than from random mini-batch ordering. Thus, when evaluating with pre-training, we use randomly initialized *non-pre-trained* members.

**PowerMargin and PowerBALD.** Like many other AL methods, both margin and BALD were designed for the case when points are acquired one at a time (i.e. batch size 1). Recently, Kirsch et al. (2021) proposed a simple and efficient technique for extending any single sample acquisition method to the batch setting. Letting $\{s_i\}$ represent the scores for the candidate points $\mathcal{Z}$, instead of selecting $\text{topk}\,\{s_i\}$, they propose selecting $\text{topk}\,\{s_i + \epsilon_i\}$ (*softmax* variant) or $\text{topk}\,\{\log(s_i) + \epsilon_i\}$ (*power* variant), where $\epsilon_i \sim \text{Gumbel}(0, \beta^{-1})$. The driving insight, derived from the popular Softmax-Gumbel trick (Maddison et al., 2014; Gumbel, 1954; Kool et al., 2019), is that $\text{topk}\,\{s_i + \epsilon_i\}$ is the same as sampling stochastically without replacement from $\mathcal{Z}$ where the sampling distribution is Categorical $\left(\frac{\exp(\beta s_i)}{\sum_j \exp(\beta s_j)}, i \in \{1, \ldots, |\mathcal{Z}|\}\right)$. As they recommend, we use the *power* variant with $\beta = 1$ for both BALD and margin with $1 - \text{Margin}(\cdot)$ used for the latter.

## 4 EXPERIMENTS

### 4.1 SETUP

**Active Learning Setting.** We consider batch-based AL in this work. Starting with a seed set $\mathcal{I}_0$ of labeled points drawn from the training split, we select the best $B$ points—according to the examined AL method—from the remainder of training to be labeled and added to our active set at each acquisition round. We do this iteratively for $T$ rounds or until the training dataset is exhausted, whichever happens first.

In order to get a clear picture into the performance of AL algorithms across active learning settings of practical interest, we construct the following scenarios, fixing $T = 20$. **Small**: $|\mathcal{I}_0| = 30$, $B = 10$. **Medium**: $|\mathcal{I}_0| = 100$, $B = 50$. **Large**: $|\mathcal{I}_0| = 300$, $B = 200$.

**Datasets.** We consider the same 69 tabular datasets used in Bahri et al. (2021) and perform the same pre-processing steps. Concretely, these are all the datasets from the public OpenML-CC18 classification benchmark[1] under the CC-BY licence, less MNIST, Fashion-MNIST, and CIFAR10 as they are vision-centric. We pre-process as follows: if a feature column is always missing, we drop it. Otherwise, if the feature is categorical, we fill in missing entries with the mode, or most frequent, category computed over the full dataset. For numerical features, we impute it with the mean. We represent categorical features by a one-hot encoding. We z-score normalize (i.e. subtract the mean and divide by the standard deviation) all numerical features of every dataset except three (OpenML dataset ids 4134, 28, and 1468), which are left unscaled. For each OpenML dataset, we form $80\%/20\%$ train/test splits where a different split is generated for each of the 20 trials and all methods use the same splits. Unsupervised SCARF pre-training uses the features (and not the labels) of the entire train split – $70\%$ for training and the remaining $10\%$ as a validation set for early stopping.

**Model Architectures and Training.** Our model consists of a backbone followed by a classification head (a single affine projection down to number of classes). The backbone is a 5-layer ReLU deep net with 256 units per layer. When the model is SCARF pre-trained, a pre-training head, a 2-layer ReLU net with 256 units per layer, is attached to the output of the backbone. After pre-training the backbone with the pre-training head, the head is discarded; both the backbone and the classification head are updated during supervised fine-tuning. We use the recommended settings for SCARF – $60\%$ of the feature indices are corrupted and no temperature scaling (i.e. $\tau = 1$). We pre-train for a maximum of 1000 epochs, early stopping with patience 3 on a static validation set built from 20 epochs of the validation data. We train all models with the Adam optimizer using default learning rate 0.001 and a batch size of 128. For supervised training, we minimize the cross-entropy loss for 30 epochs.

---

[1] https://docs.openml.org/benchmark/

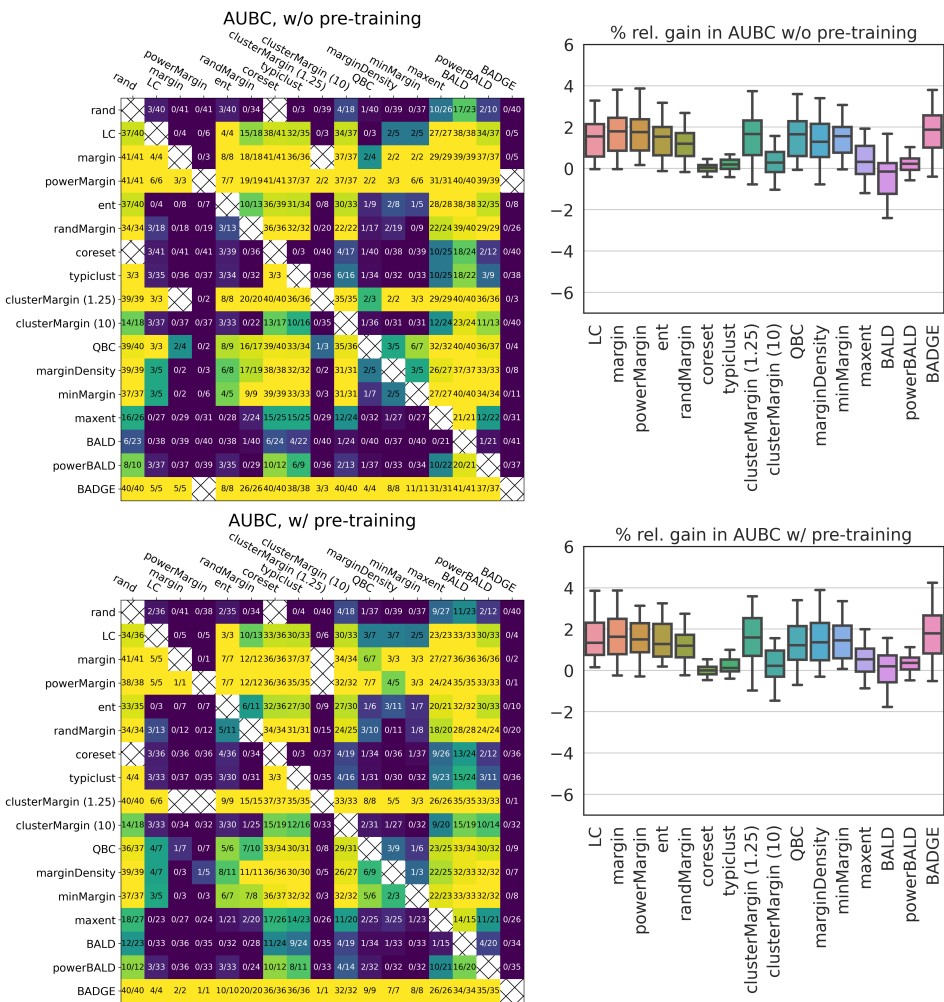

Figure 1: **Medium** scale win and box plots. The box plots shown are unfiltered (those filtered by $p$-value are shown in the Appendix). We see with and without pre-training, margin matches or outperforms alternatives on nearly all datasets for which there was a statistically significant difference. For example, without pre-training, it outperforms random all 41 of 41 times, CoreSet 41 of 41 times, and BALD 37 of 37 times. The relative gain over random is about 1-3%. See §4.2 for details on the statistical computation.

**Implementation and Infrastructure.** Methods were implemented using the Keras API of Tensorflow 2.0. Experiments were run on a cloud cluster of CPUs, and we used on the order of one million CPU core hours in total for the experiments.

## 4.2 EVALUATION METHODS

*Win matrix.* Given $M$ methods, we compute a "win" matrix $W$ of size $M \times M$, where the $(i,j)$ entry is defined as:

$$W_{i,j} = \frac{\sum_{d=1}^{69} w(i,j,d)}{\sum_{d=1}^{69} w(i,j,d) + l(i,j,d)}, \text{ where}$$

$$w(i,j,d) = \mathbb{1}[\text{method } i \text{ beats } j \text{ on dataset } d], \text{ and}$$

$$l(i,j,d) = \mathbb{1}[\text{method } i \text{ loses to } j \text{ on dataset } d].$$

"Beats" and "loses" are only defined when the means are not a statistical tie (using Welch's $t$-test with unequal variance and a $p$-value of 0.01). A win ratio of $0/1$ means that out of the 69 (pairwise)

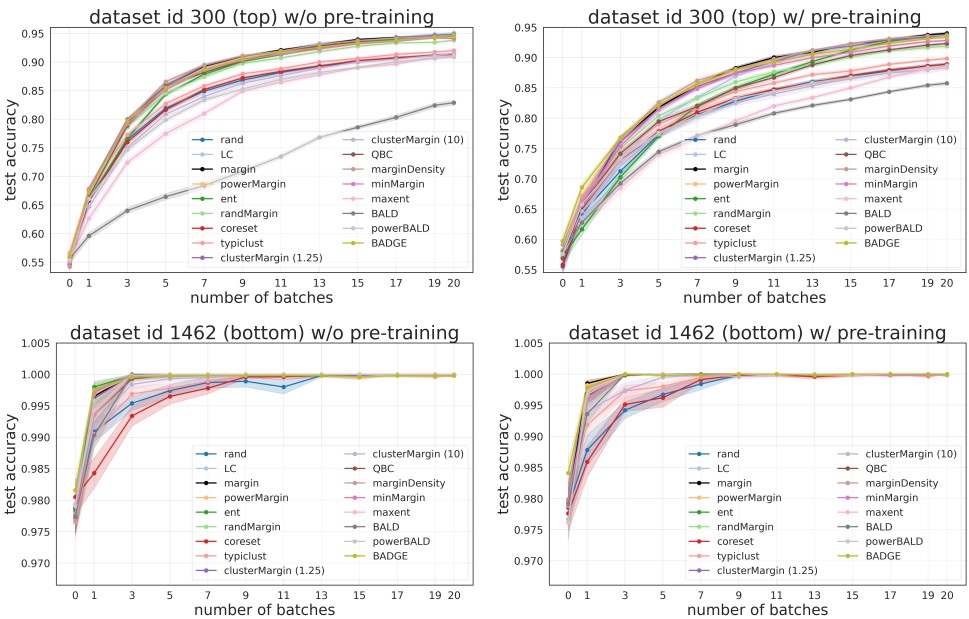

Figure 2: **Medium** scale AL curves. Margin has strong, stable performance across rounds for its best (top) and worst (bottom) datasets alike, with and without pre-training. Average dataset is in the Appendix.

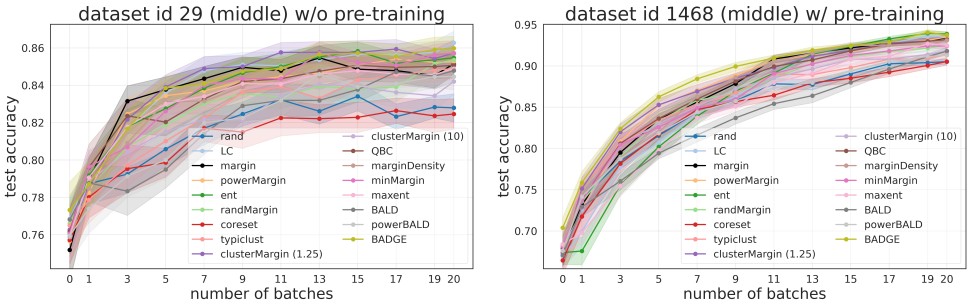

Figure 3: **Small** scale AL curves. Like the medium setting, margin has strong, stable performance across rounds for its best, average, and worst datasets alike, with and without pre-training. See Appendix for best and worst datasets.

comparisons, only one was significant and it was a loss. Since $0/1$ and $0/69$ have the same value but the latter is more confident indication that $i$ is worse than $j$, we present the values in fractional form and use a heat map.

*Box plots.* The win matrix effectively conveys how often one method beats another but does not capture the degree by which. To that end, for each method, we compute the relative percent improvement over the random sampling baseline on each dataset. We then build box-plots depicting the distribution of the relative improvement across datasets. We show relative gains on *all* datasets as well as gains only on statistically significant datasets where the means of the method and the reference are different with $p$-value $0.1$ (we use a larger $p$-value here than with the win ratio to capture more points).

We show win ratio and box plots for the Area Under the Budget Curve (AUBC) metric, where the x-axis is "number of batches acquired" and the y-axis is test accuracy. The trapezoidal rule is used to calculate the area.

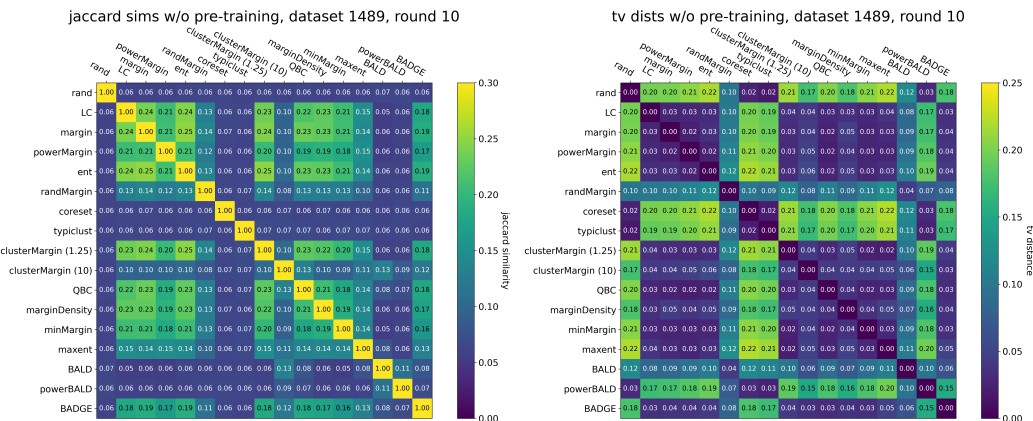

Figure 4: Pairwise comparisons of Jaccard similarities and TV distances for an example dataset and round without pre-training under the medium setting. The $(i, j)$ entry compares methods $i$ and $j$. Metrics are averaged over the number of independent trials. The standard errors are low (TV: median = 0.006, max = 0.03; Jaccard: median = 0.002, max = 0.012). Results with pre-training have the same trends (see Appendix).

*Probability of Improvement.* Following the methodology of Agarwal et al. (2021), we estimate the probability that a method will outperform margin when the dataset is unknown and we use stratified bootstrapping to estimate confidence intervals of this statistic. See the Appendix for details.

## 4.3 MEDIUM SETTING RESULTS

Figure 1 shows results for our medium scale active learning setup. Firstly, all baselines except for CoreSet, Max-Entropy, BALD, and PowerBALD methods are able to outperform the random sampling baseline on roughly half of all the datasets. Thus, actively selecting which points to label does in fact help model performance in this regime. We find that with and without pre-training, margin slightly outperforms Least Confident and Entropy (the other uncertainty baselines) along with QBC, Margin-Density and Min-Margin. It significantly outperforms clustering-centric baselines TypiClust and Cluster-Margin, if the latter uses an average cluster size of 10. When Cluster-Margin uses an average cluster size of 1.25, performance is similar to margin. This makes sense, since in this case, the algorithm first selects the roughly $B$ lowest margin points, and each point will more or less have its own cluster (except perhaps if two datapoints are near duplicates), so sampling from these clusters will just return the same lowest margin points. From the box plots we see that margin beats random by a relative 1-3% on average. Methods that tied or slightly underperformed margin on the win plots have a comparable relative gain over random, whereas the gains for others are near zero or negative (in the case of BALD). For each pre-training setting, Figure 2 shows AL curves for the datasets margin performs best, average, and worst on, compared to random (filtering using a $p$-value of 0.1). We clearly see margin near or at the top of the pack in all cases and across all acquisition rounds.

## 4.4 SMALL SETTING RESULTS

Small setting results are shown in Figures 3, 11, 6, 10 (the last 3 are in the Appendix). Somewhat to our surprise, the trends in the small setup are similar to those for the medium scale setup. A priori it seemed that with a seed set size of merely 30, embeddings or uncertainties derived from the current model would be untrustworthy and that random sampling would often be optimal. However, we find that even in this regime, margin and other uncertainty-based baselines provide a boost. We see that as before, CoreSet, TypiClust, PowerBALD, and Max-Entropy perform similar to random while BALD underperforms substantially. For example, without pre-training, margin outperforms random all 39 of 40 times, CoreSet 36 of 36 times, and BALD 44 of 44 times. The relative gain over random is about 1-4%.

### 4.5 LARGE SETTING RESULTS

In this setting we use a larger seed set and a larger batch, to test whether starting with a more accurate underlying model will benefit alternatives more than margin and whether margin's naive top-k batch acquisition approach would be overshadowed by the other baselines' diversity-promoting ones. With the caveat that what sizes are considered small or large is subjective and application specific, we observe no differences in high level trends in this setting. Results are presented in Figures 7, 8, and 9, shown in the Appendix.

### 4.6 DEEPER ANALYSIS

Our analysis thus far has only looked at each method's overall test accuracy. While it can be challenging to pinpoint *why* one method outperforms another, we attempt to provide some insight by comparing which examples each method chooses to acquire. Specifically, we track which examples (i.e. the "example ids") comprise the active set after each round. We discard the seed set since it is shared across all methods and it dilutes the metrics. Like the win plots, we do pairwise comparisons. Given some dataset, let $E[i, r]$ be the example ids in the active set for method $i$ after round $r$ (i.e. examples acquired at or before round $r$) and let $P[i, r]$ be the probability distribution over class labels for such examples. We use the Jaccard similarity to measure the degree of overlap in acquired examples between two methods and the Total Variation (TV) distance between the class label distributions of these examples:

$$\text{Jaccard}(i, j, r) = \frac{|E[i, r] \cap E[j, r]|}{|E[i, r] \cup E[j, r]|}$$
$$\text{TV}(i, j, r) = \max_k |P[i, r][k] - P[j, r][k]|.$$

Figure 4 depicts the pairwise comparisons on an example dataset and round for the medium setting. Without pre-training, we see that the Jaccard similarity with random is about 0.06, so this serves as a "uncorrelated" baseline score that captures overlap caused only because of the finite size of the training set. Scores above this indicate positive correlation in how the methods choose to acquire samples. Interestingly, we see that (1) Least Confident, PowerMargin, Entropy, Cluster Margin (1.25), QBC, Margin-Density, Min-Margin have strong (around 0.23) overlap with margin, (2) Random-Margin, Max-Entropy, and BADGE have moderate overlap, and (3) CoreSet, TypiClust, Cluster Margin (10), BALD, and PowerBALD have little-to-no noteworthy overlap. This suggests that margin's uncertainty-based alternatives operate in a similar way (i.e. low margin samples may also be low confidence and high entropy points), whereas clustering-based methods do not, especially as fewer clusters are used.

Meanwhile, the TV distance plots indicate the following by way of comparison against random (which should preserve the underlying class distribution). (1) CoreSet, TypiClust, PowerBALD are very close to random (around 0.02) (2) BADGE, BALD, Random-Margin are moderately close to random (around 0.10) and (3) the remaining methods are far from random. This indicates that (1) margin and other uncertainty methods do not preserve class balance as they acquire samples, whereas clustering-based ones do, and (2) maintaining class balance is not vital (and sometimes sub-optimal) for achieving good performance in active learning.

## 5 CONCLUSION

In this work, we question whether many active learning strategies, new and old alike, can really outperform simple margin sampling when deep neural networks are trained on small to medium-sized tabular datasets. We analyzed a diverse set of methods on 69 real-world datasets with and without pre-training under different seed set and batch sizes, and we found that no method was able to outperform margin sampling in any statistically remarkable way. Margin has no hyper-parameters and is consistently strong across all settings explored. We cannot recommend a single better method for practitioners faced with data labeling constraints, especially in the case of tabular data, or baseline to be benchmarked against by researchers in the field.

## 6 REPRODUCIBILITY

We strive to make our findings completely reproducible. The datasets we use are open-source and all data processing steps and train/test split constructions are carefully detailed (see Section 4). All aspects of our model architecture and training are also described elaborately, as are all baseline methods – both the methods themselves and every hyper-parameter used. The goal of this work is to provide a careful and statistically sound benchmark of active learning methods across a large number of tabular datasets; to that end, we have presented our results through visual devices like box plots, win matrices, line plots, and probability of improvement plots, all of which include measures of certainty (e.g. $p$-values). Our goal with this work is to make a very convincing point that margin matches or outperforms alternative active learning techniques for tabular data and we believe we have included every detail necessary for anyone to recreate our results.

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

# A APPENDIX

We present results that were omitted from the main text.

## A.1 PROBABILITY IMPROVEMENT: AN ALTERNATIVE TO WIN RATIO PLOTS

Agarwal et al. (2021) suggests various statistically sound strategies for comparing the performance of methods across tasks (or datasets) in the presence of stochastic factors. One such strategy is *probability of improvement*, which we briefly review for completeness (see the paper for more details).

Let $X$ and $Y$ be the scalar performance metric (higher is better) of algorithms $X$ and $Y$, and $X_m$ ($Y_m$) be the performance of $X$ ($Y$) on dataset $m$. Suppose we observe $N$ samples of $X_m$ and $K$ samples of $Y_m$. We use the Mann-Whitney U-statistic:

$$P(X_m > Y_m) = \frac{1}{NK} \sum_{i=1}^{N} \sum_{j=1}^{K} S(x_{m,i}, y_{m,j}) \quad \text{where} \quad S(x,y) = \begin{cases} 1, & \text{if } y < x, \\ \frac{1}{2}, & \text{if } y = x, \\ 0, & \text{if } y > x. \end{cases}$$

$$P(X > Y) = \frac{1}{M} \sum_{i=1}^{M} P(X_m > Y_m),$$

where $x_{m,i}$ represents the performance of $X$ on trial $i$ on dataset $m$. We perform stratified bootstrap sampling (re-sampling 200 times from $X_{m,1:N}$ and $Y_{m,1:K}$ for each dataset $m$) and then show violin plots of the bootstrap sampling distribution of the U-statistic (i.e. probability of improvement). If the upper CI is higher than a threshold of 0.75, then the results are said to be statistically meaningful as per the Neyman-Pearson statistical testing criterion.

In Figure 5 we plot the probability that a method beats margin on the AUBC metric. We find that this probability is less than around 0.50 and the quantiles are well below 0.75 so we can say that when the dataset / task is unknown, no method outperforms margin in any reasonably statistically meaningful way.

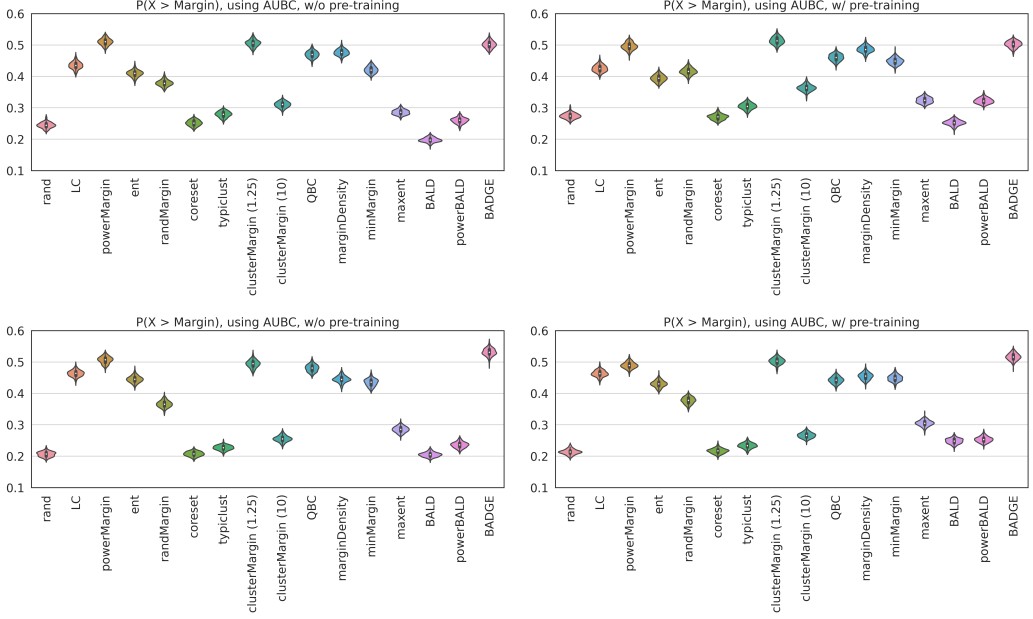

Figure 5: Probability of Improvement charts for **small** (top) and **medium** (bottom) settings. We see that the alternatives to margin do not beat margin in a statistically meaningful way.

## A.2 SMALL SETTING RESULTS

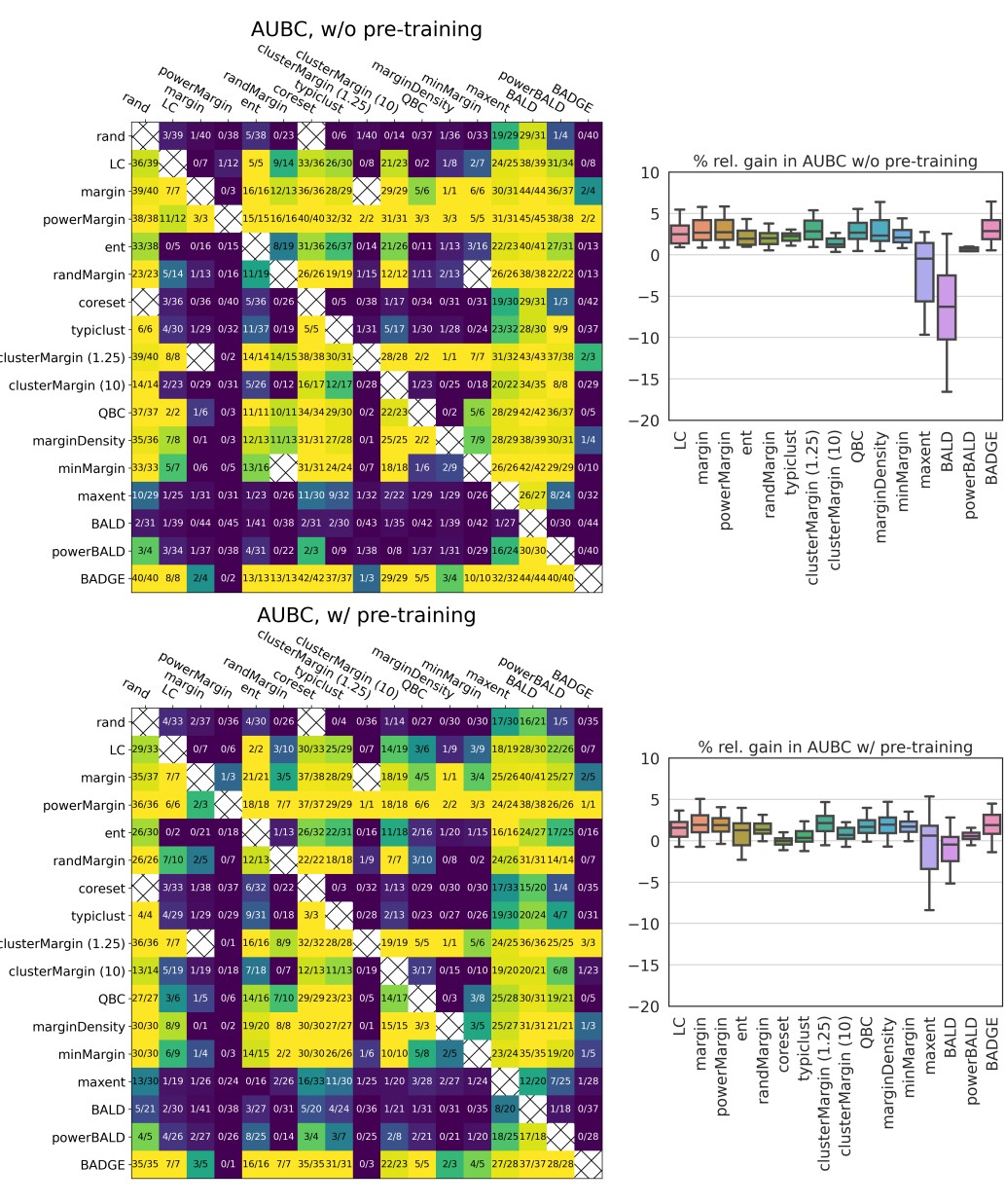

Figure 6: Win and unfiltered box plots for the **small** setting.

## A.3 LARGE SETTING RESULTS

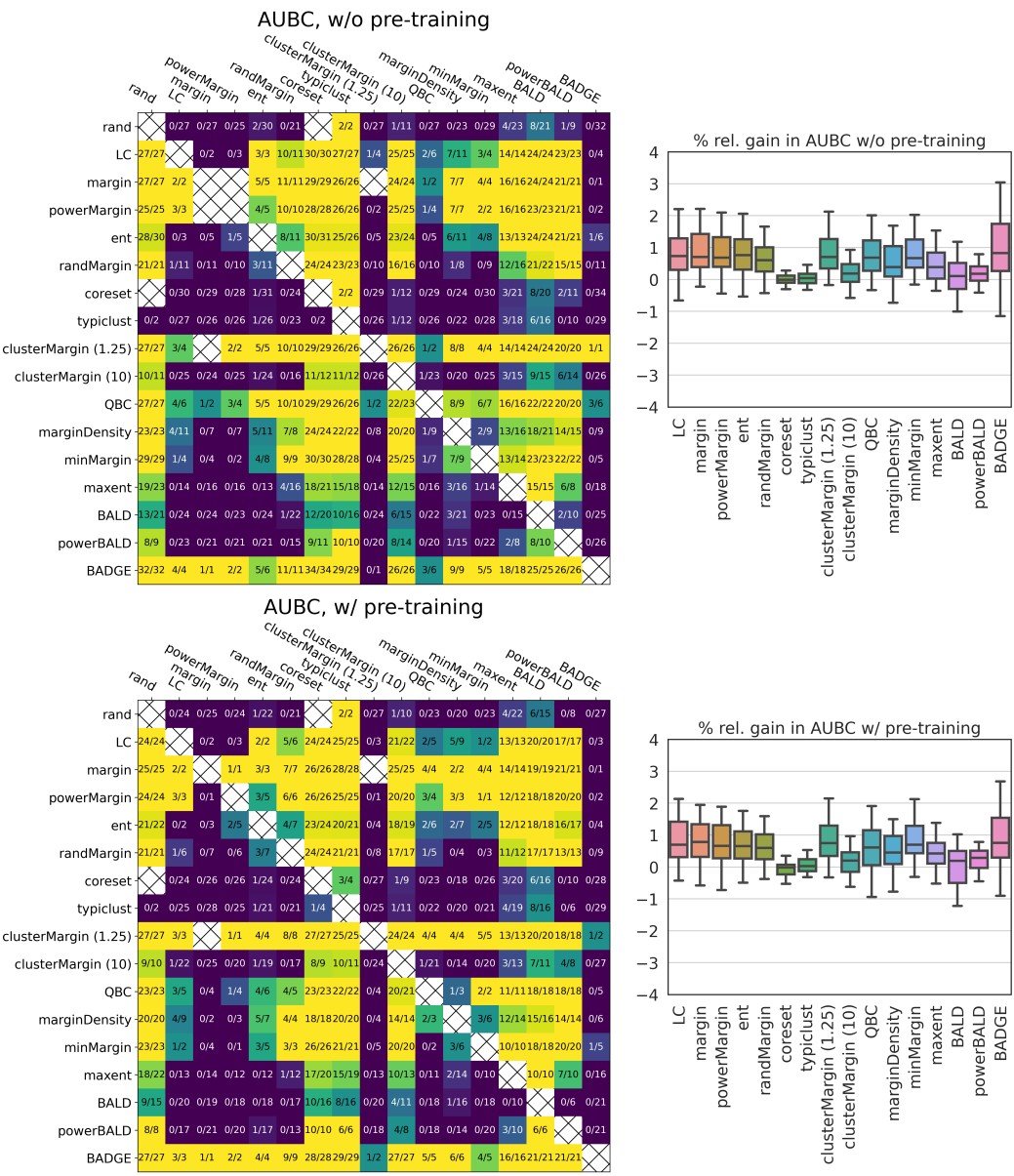

Figure 7: Win and unfiltered box plots for the **large** setting.

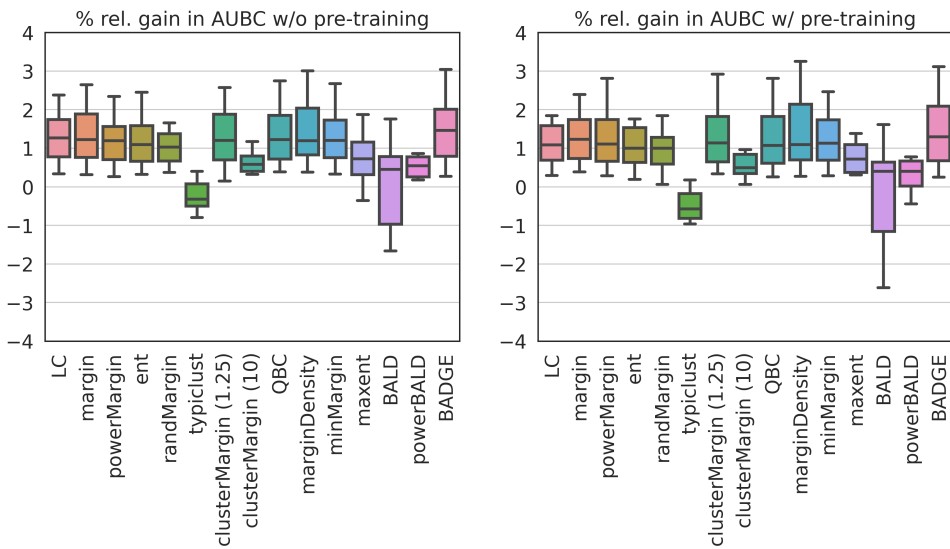

Figure 8: Box plots for the **large** setting filtered using a $p$-value of 0.1.

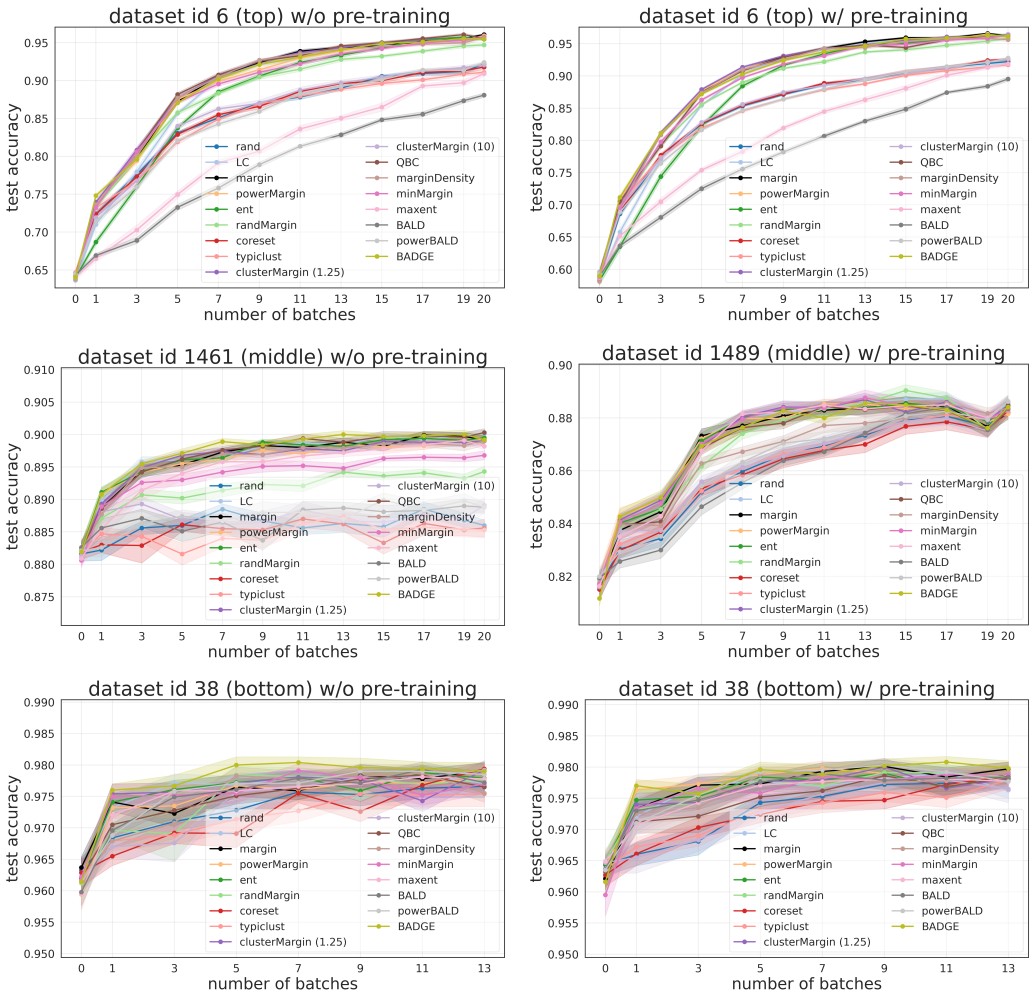

Figure 9: AL curves for the **large** setting.

## A.4 SMALL AND MEDIUM SCALE RESULTS

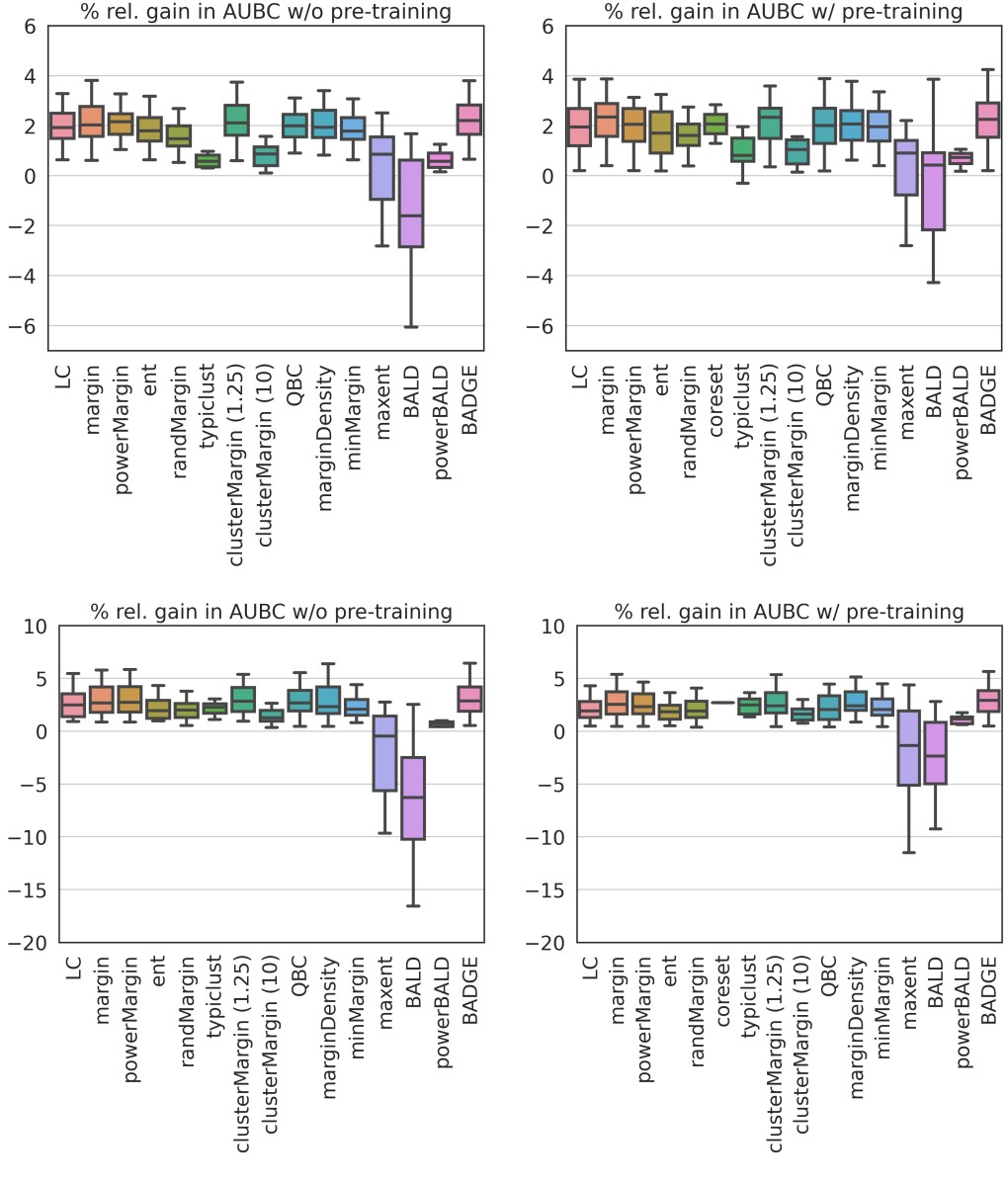

Figure 10: **Top**: Box plots for the **medium** setting filtered using $p$-value 0.1. **Bottom**: Same plots for the **small** setting.

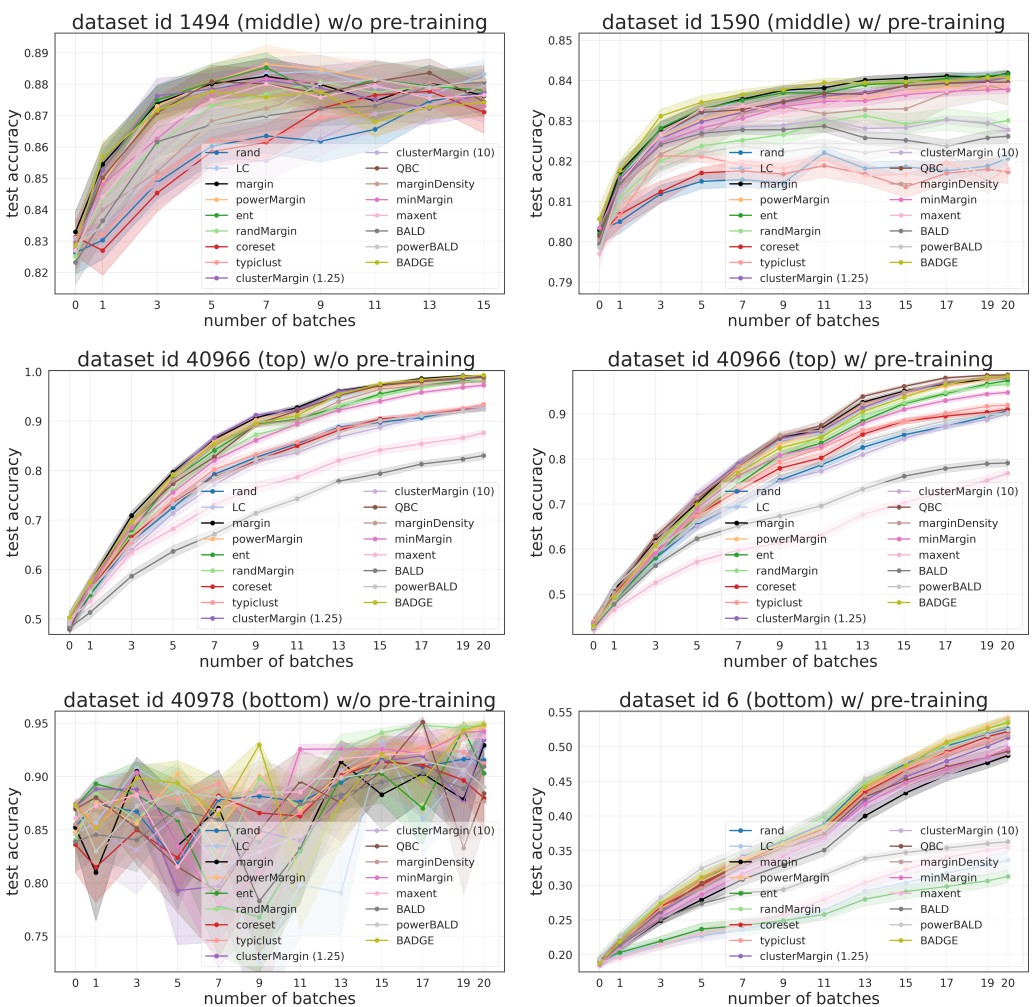

Figure 11: AL curves: middle curves for **medium** setting (top), top and bottom curves for the **small** setting (bottom two).

## A.5 JACCARD SIMILARITY AND TV DISTANCE PLOTS

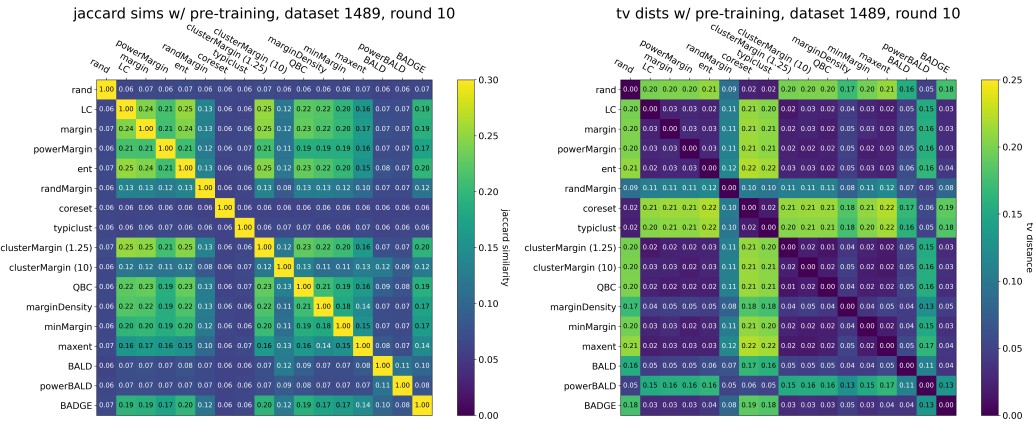

Figure 12: Pairwise comparisons of Jaccard similarities and TV distances for an example dataset and round with pre-training under the medium setting.

