# OpenReview forum: "Is margin all you need? An extensive empirical study of deep active learning on tabular data"
_ICLR.cc/2024/Conference — Submitted to ICLR 2024_

### Official Review · Reviewer_rdp2 · 2023-10-31

**Soundness:** 3 good
**Presentation:** 3 good
**Contribution:** 2 fair
**Rating:** 3
**Confidence:** 2

**Summary:**

This paper conducts an empirical study on various active learning algorithms on deep neural networks trained on 69 tabular classification datasets. The results show that the margin-based sampling techniques can achieve good performance.

**Strengths:**

The paper conducts extensive experiments on tabular data, the results show that the margin-based sampling techniques can achieve comparable performance. This can inspire researchers to consider this problem.

**Weaknesses:**

1. There lack of analysis about the experimental results.
2. Actually, deep neural networks can not achieve good performance on tabular data. This may influence the fairness of the experiments.

**Questions:**

Deep neural networks can not achieve good performance on tabular data. Will this influence the fairness of the experiments? How about the performance of different sampling techniques on other data types, such as image?

---

> ### Author Response · Authors · 2023-11-21
> **rebuttal**
>
> As we note in the main text, the SCARF paper compares deep NNs to gradient boosted decision trees and shows that on the contrary, deep NNs are competitive (especially when pre-trained). Could you please post experimental results or references that contradict this?
>
> The scope of the paper is tabular data, as we explicitly state quite a few times. Others have studied images and text, as we mention in the Related Works: "There have also been empirical evaluations of active learning procedures in the non-tabular case. Hu et al. (2021) showed that margin attained the best average performance of the baselines tested on two image and three text classification tasks across a variety of neural network architectures and labeling budgets."

---

### Official Review · Reviewer_uF25 · 2023-10-31

**Soundness:** 2 fair
**Presentation:** 3 good
**Contribution:** 2 fair
**Rating:** 5
**Confidence:** 5

**Summary:**

This paper conducts a comprehensive benchmark of many active learning techniques on tabular data. The experiments are tested on basic learners both with and without pretraining on the data. The study encompasses a substantial array of datasets sourced from OpenML. The key observations are that the classic margin-based method frequently demonstrates competitive performance, often not significantly worse than state-of-the-art approaches. Surprisingly, some methods like BALD exhibit significantly worse performance than random selection in this context.

**Strengths:**

- This paper is well-written and easy to follow.

- This paper is skillfully composed and presents an extensive comparative survey that encompasses a wide range of 69 tabular datasets.

**Weaknesses:**

- The conclusion section states that margin sampling outperformed other strategies in deep neural network models. However, it is important to note that the experiment solely relied on a specific model architecture (SCARF) as the backbone. Consequently, it is recommended to diversify the model architectures used in the experiments to ensure the generalizability of the findings.

- It is advisable to incorporate an ablation study and detailed analysis to assess the potential influence of pre-training backbones on the performance of various active learning strategies, including margin sampling.  This additional analysis would provide valuable insights into the influence of pre-training on the efficacy of different AL methods.

- To enhance the persuasiveness of the conclusions, it is crucial to conduct more experiments encompassing a broader range of active learning settings. The current results are derived from a limited set of active learning settings, and expanding this scope would enhance the overall validity and generalizability of the study's findings.

- For the results to be truly groundbreaking and instructive, the paper should strive for innovation and provide additional theoretical analysis and in-depth discussions. While the current implication is that margin sampling excels as a general active learning strategy, it would be significantly enhanced by offering a deeper understanding of why margin sampling outperforms other methods. Moreover, practical suggestions should be included to guide future research in selecting the most appropriate active learning techniques for specific scenarios.

**Questions:**

See above.

---

> ### Author Response · Authors · 2023-11-21
> **rebuttal**
>
> Thank you for the review.
>
> To clarify, the model architecture is a feedforward ReLU network. SCARF is an architecture agnostic pre-training algorithm. We show results with and without SCARF pre-training. Since our domain is tabular data, within the realm of deep models, feedforward networks are the standard choice (though niche alternatives exist) -- which is to say, we don't think it'd make much sense to try CNNs or Transformers here.
> Our paper precisely ablates the effect of SCARF pre-training in every experiment. SCARF is the most appropriate pre-training scheme to consider for this paper as it is agnostic to modality and was shown to work well precisely for the type of real-world tabular data we use here.
>
> *RE: "To enhance the persuasiveness of the conclusions, it is crucial to conduct more experiments encompassing a broader range of active learning settings."*. We believe the analysis is pretty thorough; could you suggest specifically which settings are missing?
>
> *RE: "...the paper should strive for innovation and provide additional theoretical analysis and in-depth discussions."* We note in paragraph 3 of the Related Works that AL methods like Margin can be difficult to analyze theoretically and that there is little theoretical understanding of the method. We attempt to provide a deeper understanding in the "Deeper Analysis" section -- we say, "While it can be challenging to pinpoint why one method outperforms another, we attempt to provide some insight by comparing which examples each method chooses to acquire."
> We think the guidance this paper provides is clear -- practitioners doing active learning on tabular data should opt for Margin sampling.

---

### Official Review · Reviewer_eSnA · 2023-11-01

**Soundness:** 2 fair
**Presentation:** 3 good
**Contribution:** 2 fair
**Rating:** 5
**Confidence:** 4

**Summary:**

This comprehensive study analyzes the performance of a variety of AL algorithms on deep neural networks trained on 69 real-world tabular classification datasets from the OpenML-CC18 benchmark. This study finds that the classical margin sampling technique matches or outperforms all others, including current state-of-art, in a wide range of experimental settings.

**Strengths:**

Novelty: This work analyzed a diverse set of methods on 69 real-world datasets with and without pre-training under different seed set and batch sizes, which is not conducted by previous work.
Quality: Through the experiment, this paper proposes that no method is able to outperform margin sampling in any statistically remarkable way.
Clarity: This paper describes the experiment in detail, which makes it easy to follow.
Significance: This work gives the conclusion that margin has no hyper-parameters and is consistently strong across all settings explored, which proves it safe to commit to margin sampling for practitioners.

**Weaknesses:**

1. The figures in this paper cannot prove the conclusion strongly, which is unclear.
2. BADGE outperforms all methods when existing statistically remarkable difference between them, while this paper do not analyze the reason, which makes the conclusion lacking of convince.

**Questions:**

1. What is the criteria for choosing comparison methods? Margin, Entropy and Least Confidence are all based on uncertainty, which are somewhat repeatedly. As far as I know, there are other kinds of deep active learning methods. Adding them in it will improve the convince of this work.

---

> ### Author Response · Authors · 2023-11-21
> **rebuttal**
>
> Thank you for the review.
>
> We benchmarked 16(!) diverse AL methods that are popular or purported as state-of-art. We believe these baselines provide great coverage.
>
> *RE: "As far as I know, there are other kinds of deep active learning methods"*. Could you please specify precisely which ones and why you think it is worthwhile to include them?
>
> *RE: BADGE*. Indeed, BADGE performs well in our experiments. However, looking at both the win and relative improvement plots in Figure 1, the difference with Margin is not too significant in our opinion. We would like to echo what Andreas Kirsch (a public reviewer) noted about BADGE. While Margin is linear as it involves doing a single forward pass for unlabeled points and sorting them by a single scalar value (the margin), BADGE requires a forward pass to infer labels and a backward pass to compute loss gradients for each unlabeled sample. It then needs to cluster these gradients using k-means, which typically takes $O(nkdi)$, where $n$ is the number of $d$-dimensional vectors ($d$ is the dimensionality of the penultimate layer embedding), $k$ is the number of samples to acquire (i.e. number of clusters), and $i$ is the number of iterations needed to converge. $i$ can vary greatly in practice, and can make the effective runtime of k-means either linear or super-polynomial.
> Given the little performance difference between Margin and BADGE but the significant additional computational overhead of the latter, our recommendation to practitioners remains Margin. We will update the paper with discussion on this point.

---

### Official Review · Reviewer_vfAb · 2023-11-05

**Soundness:** 3 good
**Presentation:** 3 good
**Contribution:** 3 good
**Rating:** 6
**Confidence:** 2

**Summary:**

-This paper examines active learning (AL) methodologies within the context of training deep neural network on tabular datasets. It evaluates various AL algorithms on 69 real-world tabular classification datasets from the OpenML-CC18 benchmark. This evaluation encompasses considerations of varying data conditions and the implications of self-supervised model pre-training. The results of the study reveal that the conventional margin sampling technique consistently demonstrates comparable or superior performance in comparison to alternative AL methods, including the most current state-of-the-art methodologies, across a spectrum of experimental configurations. It is worth noting that margin sampling is a hyperparameter-free approach, making it a robust and advantageous choice for practitioners dealing with data labeling constraints, especially for tabular data. This paper suggests that margin sampling should be recognized as both a benchmark for research investigations and a practical strategy for practitioners.

**Strengths:**

-The paper conducts a thorough and comprehensive study of AL algorithms using real-world tabular classification datasets. It rigorously compares various AL algorithms, encompassing both traditional and state-of-the-art approaches, against the classical margin sampling technique. This comparative analysis effectively highlights the relative strengths and weaknesses of these methods. The results contribute to a comprehensive understanding of AL method performance in various settings and may help offer valuable insights to assist researchers and practitioners in selecting the most suitable AL algorithms for their specific problems.

-The finding that margin sampling consistently matches or outperforms other AL strategies across a wide range of experimental settings is also an advantage, underscoring the robust and dependable nature of the simple margin sampling method for practitioners.

**Weaknesses:**

-The paper assesses various AL algorithms within the framework of tabular datasets. It's important to note that the outcomes may not be universally applicable, as the study does not investigate their potential implications in other contexts or domains, such as image data and others, potentially limiting its broader relevance.

-While the paper underscores the effectiveness of margin sampling, it falls short in offering comprehensive practical guidelines and actionable recommendations for practitioners seeking to make informed choices regarding AL algorithms in real-world applications.

**Questions:**

-How might the paper be enhanced to investigate the transferability of the studied AL algorithms to diverse domains, beyond tabular datasets, to provide a more comprehensive understanding of their applicability across different contexts? Also, it may be very helpful if the paper can provide more comprehensive and practical guidance to assist practitioners in effectively selecting and implementing AL algorithms in real-world applications.

---

> ### Author Response · Authors · 2023-11-21
> **rebuttal**
>
> Thank you for the detailed review.
>
> The focus of this work is a large-scale analysis specifically on real-world tabular data. We think this is much needed as a lot of the other active learning papers have focused solely on images or text. We have good evidence that Margin works well in these other domains. For example, in paragraph 2 of the Related Works, we note that "Hu et al. (2021) showed that margin attained the best average performance of the baselines tested on two image and three text classification tasks across a variety of neural network architectures and labeling budgets."
>
> The actionable recommendation the paper is making for practitioners is precisely that they should use Margin sampling: "Margin has no hyper-parameters and is consistently strong across all settings explored. We cannot recommend a single better method for practitioners faced with data labeling constraints, especially in the case of tabular data, or baseline to be benchmarked against by researchers in the field."

---

### Public Comment · ~Andreas_Kirsch1 · 2023-11-17
**Some comments**

Great paper! I would recommend streamlining and polishing the diagrams (the legend is not sorted), and please add more concrete recommendations to the introduction and conclusion. That is it would be great if the plot legend was sorted by average performance and the win matrices and box plots were sorted by overall number of wins or similar to make it easier to read off the best methods.

Re conclusion: BADGE seems very strong as well. Maybe it would be good to compare the compute requirements for BADGE vs margin/powerMargin in the conclusion/recommendations because one is quadratic (BADGE) and the other linear (*Margin). Having a table (or plot) with the runtime costs of the different acquisition methods might also provide valuable information in general.

Personally and obviously, I'm very happy to see that powerMargin works very well across the different sampling regimes and in comparison to many other acquisition methods.

The paper seems empirically sound and will be useful for knowing what to try when working with different datasets and acquisition sizes.

Thanks,
 Andreas

---

> ### Author Response · Authors · 2023-11-21
> **response**
>
> Andreas, we really appreciate your constructive feedback!
>
> Sorting the legends in the line plots and the order of methods in the win matrices & box plots are great suggestions and we will try to do this for the final version.
>
> You make a great point about Margin's runtime advantage over BADGE. We generally did not consider runtime, but given this paper is targeting practitioners in the real world, some discussion around this is apt. Reporting wall-clock times can often be misleading because of a strong dependence on implementation. In the final version, we will add a discussion on the runtime complexities.

---

### Author Response · Authors · 2023-11-21
**rebuttal**

Thank you all for the feedback! We address reviews individually.

---

### Meta-Review · Area_Chair_QZBD · 2023-12-08

**Metareview:**

This paper tries to present an extensive empirical comparison of active learning approaches. There are several recent papers which attempt the same. The reviewers have pointed several critical issues including, (i) the lack of diversity of domains of evaluating, (ii) the lack of conclusions and analysis of the results, lack of practical guidelines and recommendations, and so on.

In summary, this paper does not stand out given that it is a comparison of AL approaches. I would encourage the authors to consider the reviewers suggestions and submit this to another venue.

**Justification For Why Not Higher Score:**

This paper does not meet the bar required for acceptance for a evaluation paper. While a very good start, I would encourage the authors to consider the reviewers suggestions in making a stronger and more impactful version.

**Justification For Why Not Lower Score:**

N/A

---

### Decision · Program_Chairs · 2024-01-16

Reject